# Fungi and bacteria occupy distinct spatial niches within carious dentin

Rosalyn M. Sulyanto[1,2☺], Clifford J. Beall [3☺]*, Kasey Ha[1,2], Joseph Montesano[4‡], Jason Juang[4‡], John R. Dickson[5], Shahr B. Hashmi[3], Seth Bradbury[3,6], Eugene J. Leys[3], Mira Edgerton[7], Sunita P. Ho[8], Ann L. Griffen[3,6]

1 Department of Dentistry, Boston Children's Hospital, Boston, Massachusetts, United States of America, 2 Department of Developmental Biology, Harvard School of Dental Medicine, Boston, Massachusetts, United States of America, 3 Division of Biosciences, College of Dentistry, The Ohio State University, Columbus, Ohio, United States of America, 4 Harvard School of Dental Medicine, Boston, Massachusetts, United States of America, 5 Department of Neurology, Massachusetts General Hospital and Harvard Medical School, Boston, Massachusetts, United States of America, 6 Division of Pediatric Dentistry, College of Dentistry, The Ohio State University, Columbus, Ohio, United States of America, 7 Department of Oral Biology, School of Dental Medicine, University at Buffalo, Buffalo, New York, United States of America, 8 Preventive and Restorative Dental Sciences, School of Dentistry, University of California, San Francisco, San Francisco, California, United States of America

☺ These authors contributed equally to this work.
‡ JM and JJ also contributed equally to this work.
* beall.3@osu.edu

**Data Availability Statement:** The sequence and metadata underlying this work is available in the NCBI SRA/BioProject/BioSample repositories with accession PRJNA1003388. https://www.ncbi.nlm.nih.gov/bioproject/PRJNA1003388/. The QPCR

## Abstract

The role of bacteria in the etiology of dental caries is long established, while the role of fungi has only recently gained more attention. The microbial invasion of dentin in advanced caries especially merits additional research. We evaluated the fungal and bacterial community composition and spatial distribution within carious dentin. Amplicon 16S rRNA gene sequencing together with quantitative PCR was used to profile bacterial and fungal species in caries-free children (n = 43) and 4 stages of caries progression from children with severe early childhood caries (n = 32). Additionally, healthy (n = 10) and carious (n = 10) primary teeth were decalcified, sectioned, and stained with Grocott's methenamine silver, periodic acid Schiff (PAS) and calcofluor white (CW) for fungi. Immunolocalization was also performed using antibodies against fungal β-D-glucan, gram-positive bacterial lipoteichoic acid, gram-negative endotoxin, *Streptococcus mutans*, and *Candida albicans*. We also performed field emission scanning electron microscopy (FESEM) to visualize fungi and bacteria within carious dentinal tubules. Bacterial communities observed included a high abundance of *S. mutans* and the *Veillonella parvula* group, as expected. There was a higher ratio of fungi to bacteria in dentin-involved lesions compared to less severe lesions with frequent preponderance of *C. albicans*, *C. dubliniensis*, and in one case *C. tropicalis*. Grocott's silver, PAS, CW and immunohistochemistry (IHC) demonstrated the presence of fungi within carious dentinal tubules. Multiplex IHC revealed that fungi, gram-negative, and gram-positive bacteria primarily occupied separate dentinal tubules, with rare instances of colocalization. Similar findings were observed with multiplex immunofluorescence using anti-*S. mutans* and anti-*C. albicans* antibodies. Electron microscopy showed monomorphic bacterial and fungal biofilms within distinct dentin tubules. We demonstrate a previously unrecognized

data is stored with the analysis code at https://github.com/cliffbeall/dentin_caries. Other data such as micrographs are included in the manuscript figures.

**Funding:** The author(s) received no specific funding for this work.

**Competing interests:** The authors have no competing interests relating to this work.

phenomenon in which fungi and bacteria occupy distinct spatial niches within carious dentin and seldom co-colonize. The potential significance of this phenomenon in caries progression warrants further exploration.

## Author summary

Dental decay is one of the most common diseases of humans and can result in pain and serious infections. Bacteria, including *Streptococcus mutans*, have been identified as the dominant agents of decay, although the fungus *Candida albicans* is also involved. But most research has focused on the mixed-species surface biofilms that initiate caries, and much less is known about the journey of microbes toward the soft-tissue pulp at the center of the tooth, where the serious consequences of decay occur. To our surprise, using DNA-based microbial analysis and histology, we showed that bacteria and fungi travel separately from the surface through the natural channels in the tooth, the dentinal tubules, seldom occupying the same channel. Moreover, fungi are a larger fraction of the microbes in the dentin than on the surface. This previously unrecognized phenomenon in which fungi and bacteria occupy distinct spatial niches within carious dentin and seldom co-colonize suggests that the biology of tooth invasion is different from well-studied, mixed surface biofilms that initiate decay on the surface of the tooth.

## Introduction

Dental caries is a common chronic disease of childhood and significantly impacts children's oral health, systemic health, and quality of life [1–9]. The Global Burden of Disease (GBD) 2019 estimated that 500 million children have untreated primary tooth caries.[8] Early childhood caries (ECC), defined as the presence of one or more decayed, missing, or filled tooth surface in any primary tooth in children younger than 6-years of age [4], is a particularly destructive form of the disease. Due to the aggressive nature of ECC, areas of demineralization can rapidly develop into cavitations. If left untreated, cavitations can spread to pulpal tissue leading to infection, and in severe cases, life-threatening facial space involvement.[4,10] Caries prevalence and severity have not significantly declined, despite advances in our knowledge of caries pathophysiology and the availability of preventive and treatment measures [1,8,11].

Caries pathogenesis has largely been examined in surface biofilm[12–17] rather than in dentin. This overlooked aspect of caries pathophysiology is critical to understanding the mechanism of caries progression and developing interventional strategies to limit its extension deeper into dentin and the pulp. Toothaches, periapical inflammation and abscess, as well as facial infection from tissue invasion (cellulitis) all result from microbial access to the dental pulp [10,18]. These infections originate from biofilms on the surface of the tooth, but communication of surface biofilms with the pulp is not direct. Infections usually occur before the surface breakdown exposes the pulp, and in most cases the microbes must traverse the barrier of dentin to reach the pulp [19]. The dentin tubules, which are thin channels through the solid dentin, provide a pathway to the pulp [19]. Mineralized tissues, such as dentin, have historically presented unique challenges for microbiologic and histologic research due to factors such as their hard texture and need for decalcification for analysis, which may have hampered prior studies of microbes in carious dentin. Understanding factors that contribute to caries

progression, including the role of microbes in mineralized structures, is crucial, as invasion into dentin eventually results in access to the pulp and systemic involvement.

Surface biofilm bacteria have clasically been considered the primary etiologic agent of dental caries, with the creation of an acidic environment by *S. mutans* serving as a cornerstone in disease progression [20–24]. DNA-based studies have revealed that caries-associated surface biofilms are polymicrobial, with multiple additional bacterial species associated with caries [13,25–27], as well as eukaryotes, including *C. albicans* [28]. Demineralization of enamel by acid-producing bacteria in the surface biofilm results in a cavitation, initiating the entry of microbes into dentin.

Although bacteria have been the main kingdom of microbes considered in oral health, fungi are increasingly recognized for their importance to oral health conditions [29]. Historical studies investigated the role of fungi in caries [30–37], but they were largely forgotten until recently. Contemprary studies have shown that *C. albicans* is a common commensal fungus prevalent in the oral cavity with an oral carriage rate of 18.5–40.9% in healthy individuals.[38–40] While the colonization and persistence of *Candida* in the oral mucosa, saliva, and plaque biofilm have been demonstrated, their role in dental caries remains unclear. *Candida* spp. have been isolated from biofilm overlying dental caries [28,41], but they have not consistently been observed in carious lesions [42]. A systematic review and meta-analysis associated the presence of *C. albicans* with a higher risk of ECC [12]. Carriage rates are even higher in immunocompromised individuals, enabling *C. albicans* to become an opportunistic pathogen [43]. Both synergistic and antagonistic interactions between fungi and bacteria in the oral cavity have been reported. *C. albicans* forms mixed kingdom biofilms with *S. mutans* and other *Streptococcus species* [44–46], synergizes with *S. mutans*,[47],[44,48,49] and binds *S. mutans* glycosyltransferases to its cell surface.[48,50,51] Acetic acid, a byproduct of *Veillonella* and *Lactobacilli* metabolism, inhibits the growth of *Candida*, *Aspergillus*, and *Fusarium* [28]. *S. mutans*, when co-cultured with *C. albicans*, produce more formic acid, which mitigates *C. albicans* growth [28]. These findings all suggest that fungi and bacteria have a complex interconnected role in the pathobiology of caries.

Microbial invasion deep into tooth structure with eventual access to the pulp has received less attention than surface biofilm, despite it being critical to the serious consequences of dental caries [52]. DNA-based studies have shown that the microbial community composition of carious dentin is distinct from that of surface biofilms, and contains proteolytic bacteria including *Prevotella* in addition to acid-forming bacteria [13,53]. *Candida* also express proteolytic enzymes, and multiple species have been identified in carious dentin in children using culture-based methods [54]. However, spatial visualization of microbes in carious dentin has been sparsley reported. Largely forgotten, early microscopy studies investigated fungal invasion of carious dentin tubules, reporting the growth of bud fungus "boring into" dentin tubules of tooth and colonization of dentin tubules by bacteria at the coronal aspect and fungi deeper within the dentin tubules [30–37]. More recent work has shown the penetration of *Candida* into the tubules of carious dentin in HIV-positive children [55], and penetration of tubules by *C. albicans* hyphae from the pulpal dentin surface in infected root canals [56]. A few *in vitro* studies have demonstrated the propensity of *C. albicans* penetration into dentin tubules [57–59]. But altogether these findings have received little attention as a common pathway for advancing deep caries.

To elucidate both the fungal and bacterial community composition in carious dentin in ECC, comprehensive species-level profiling was performed using amplicon sequencing on samples of carious and non-carious dentin. This work confirmed the presence of fungal species in caries and showed relatively higher abundance in dentin samples compared to surface biofilm. Subsequently, histological methods and scanning electron microscopy (SEM) were used

to confirm the presence of fungi and bacteria in carious dentin and to visualize the spatial relationship between fungal and bacterial species. These imaging studies revealed a surprising biogeographic distribution of bacteria and fungi within dentin tubules. Both fungi and bacteria-filled tubules were common, but unlike polymicrobial surface biofilms, single tubules generally harbored either fungus or bacteria, but not both.

## Results

The MiSeq sequencing of 75 subjects generated 3.7 M paired-end 16S reads and 8.7 M ITS2 reads for 171 samples for each amplicon with a minimum of 445, a maximum of 292,470 and an average of 21,732 16S and 51,249 ITS2 sequences. 2.1M 16S sequences and 5.0 M ITS2 sequences were taxonomically identified by BLAST alignment to specific databases. A total of 373 bacterial species-level and 170 fungal species-level groups were found. Data sets used for microbial community analysis are archived at the National Center for Biotechnology Information Sequence Read Archive repository (BioProject ID- PRJNA1003388).

We quantitated total bacterial and total fungal DNA with qPCR. Fig 1 shows the relative quantities of bacterial DNA (Fig 1A), fungal DNA (Fig 1B) and the ratio of fungal to bacterial DNA (Fig 1C). It is notable that the quantity of fungal DNA varies over a much greater range than the quantity of bacterial DNA. The ratio of fungal to bacterial DNA increased with increasing depth of caries lesion and was highest in carious dentin (p = 0.024, Wald test of linear mixed effects model for the log-transformed ratio).

We used the qPCR-measured total fungal and bacterial abundances together with relative abundance based on taxonomic identification counts to generate estimates of absolute abundances of each bacterial and fungal species. We then performed non-metric multidimensional scaling (NMDS) ordinations of the communities based on the Bray-Curtis dissimilarity of estimated absolute abundance for bacterial and fungal communities. The resulting plots for bacterial communities are shown in Fig 2A and for fungal communities in Fig 2B. For comparison NMDS plots based on fractional abundance without qPCR are shown in S1 Fig. The results of pairwise PERMANOVA tests are shown in S1 Table. The bacterial communities shifted in a mostly consistent direction with increasing severity of lesion, and many pairwise PERMANOVAs showed significant differences between groups. This shift is shown by the arrows indicating the position of the group centroids and the ellipses indicating 95% confidence intervals of the centroids. This shift in bacterial community with severity of lesion is consistent with earlier results [13]. The fungal communities analyzed with estimated absolute abundance showed a trend where the more severe cavitated and dentin lesions tended to be present in clusters at upper right and lower right of the NMDS plot. Bubble plots (S2A and S2B Fig) indicated that those clusters represented samples that contained high abundance of either *C. albicans* or *C. dubliniensis*. For comparison, we examined a bubble plot of the bacterial NMDS with the abundance of *S. mutans*, (S2C Fig). *S. mutans* was found in high abundance in cavitated and dentin lesions but without as much separate clustering as for the two *Candida* species in fungal communities.

To find which microbial species varied with caries lesion severity, we performed Friedman tests on species-level estimated absolute abundance data from the caries subjects, using subject as the block variable. We found 79 bacterial species varied significantly with q value < 0.05 after adjustment with the Benjamini-Hochberg false discovery rate procedure [60] and three fungal species varied significantly. The volcano plots with some of the prominent species labeled are shown in Fig 3, graphing the log2 of the fold difference between intact enamel and dentin lesions versus the negative log10 of the p value in the Friedman test (S2 and S3 Tables). In the bacteria, there was a tendency for multiple species to be higher in caries lesions. *S.*

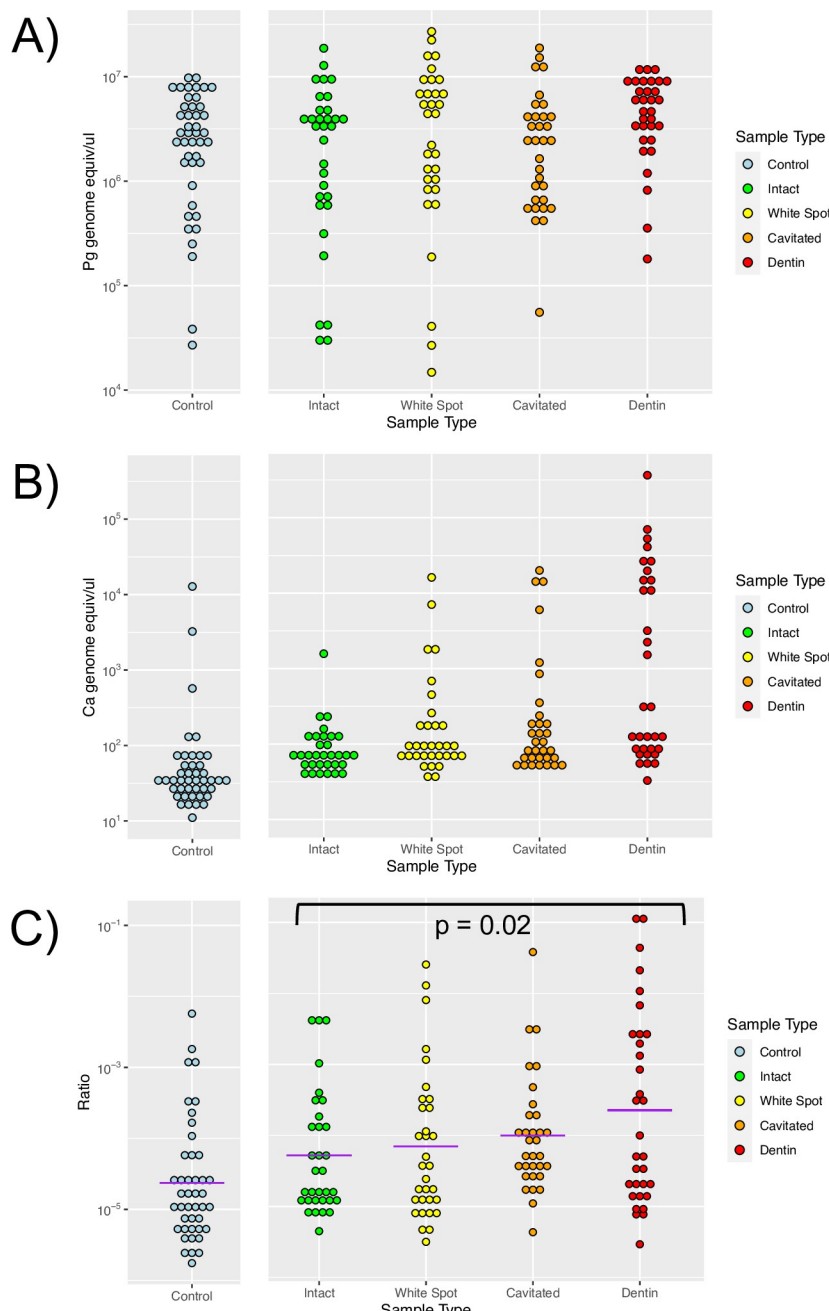

**Fig 1. Quantitation of bacterial and fungal DNA.** Total bacterial and fungal DNA was determined with qPCR using universal primers. Samples were collected from from intact enamel in control subjects and intact enamel plus white spots, cavitated, and dentin-involved lesions in caries subjects. A) Bacterial DNA quantities determined with Porphyromonas gingivalis (Pg) genomic DNA as standard B) Fungal DNA quantities determined with Candida albicans genomic DNA as standard. C) Ratio of fungal to bacterial DNA based on genome copies. The purple lines are geometric means of the groups and the p value from a linear mixed effects model of the log-transformed data for the caries subjects is shown.

*mutans* gave the lowest p value and was substantially higher in dentin lesions than in intact enamel samples. Other notable species are the *Streptococcus salivarius* group, *Prevotella histicola*, *Streptococcus parasanguinis*, *Scardovia wiggsiae*, the *Lactobacillus casei* group, and *Rothia*

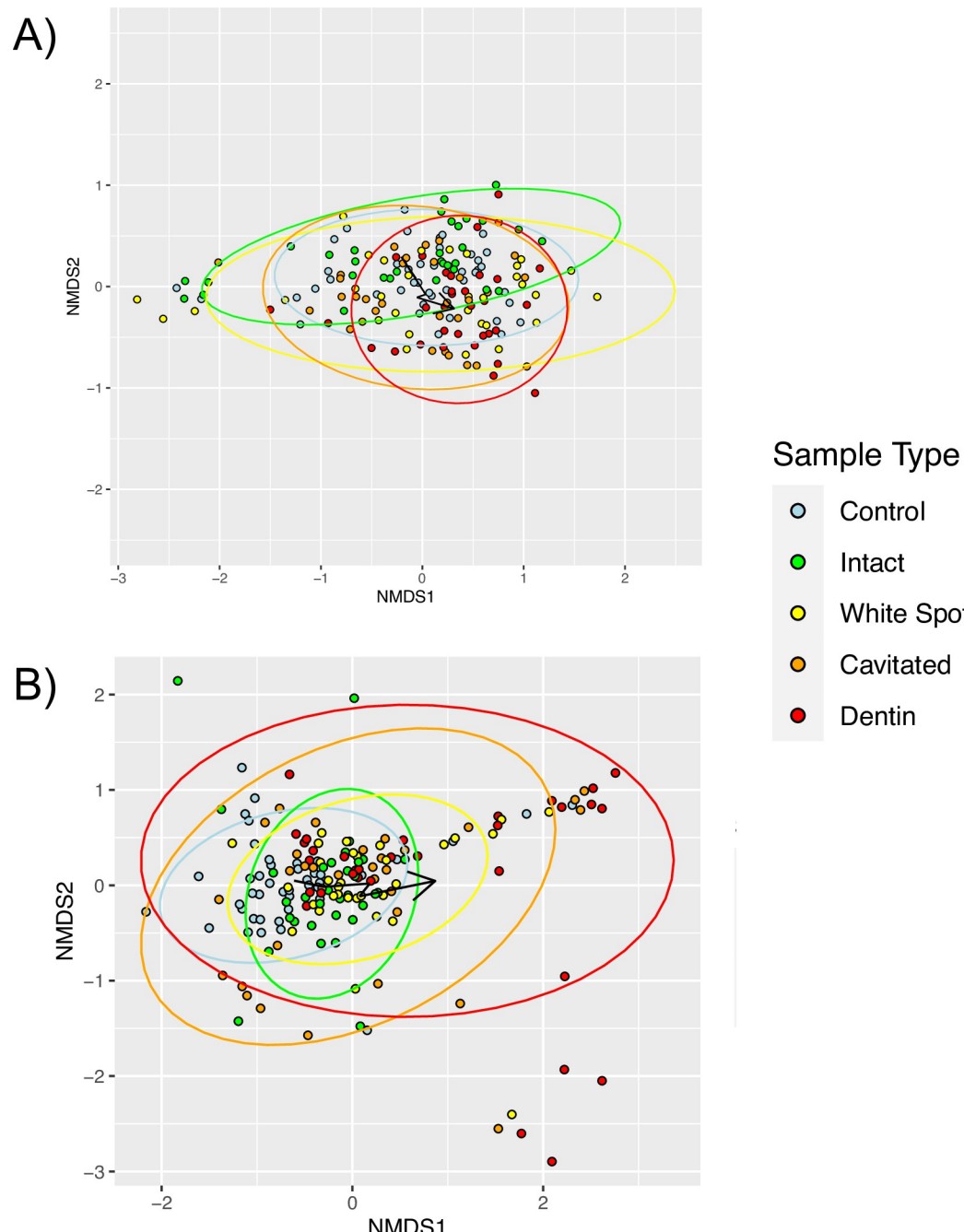

**Fig 2. Non-metric multidimensional scaling of samples based on Bray-Curtis dissimilarities of qPCR-adjusted estimate of absolute species abundances.** Arrows indicate the group centroids with increasing severity of lesion as in the legend. Ellipses are 95% confidence intervals of the centroids. A) Bacterial communities B) Fungal communities.

*mucilaginosa.* Among fungi, three *Candida* species were most notable. *C. tropicalis*, *C. dubliniensis*, and *C. albicans* were all significantly different in the Friedman test and higher in more severe lesions than in intact enamel.

We further used the absolute abundance estimate to try to predict which species might be observed in dentin lesions histologically. S3A Fig shows dot plots of the abundance per sample of the top 15 bacterial species in carious dentin and S3B Fig shows the top 5 fungal species.

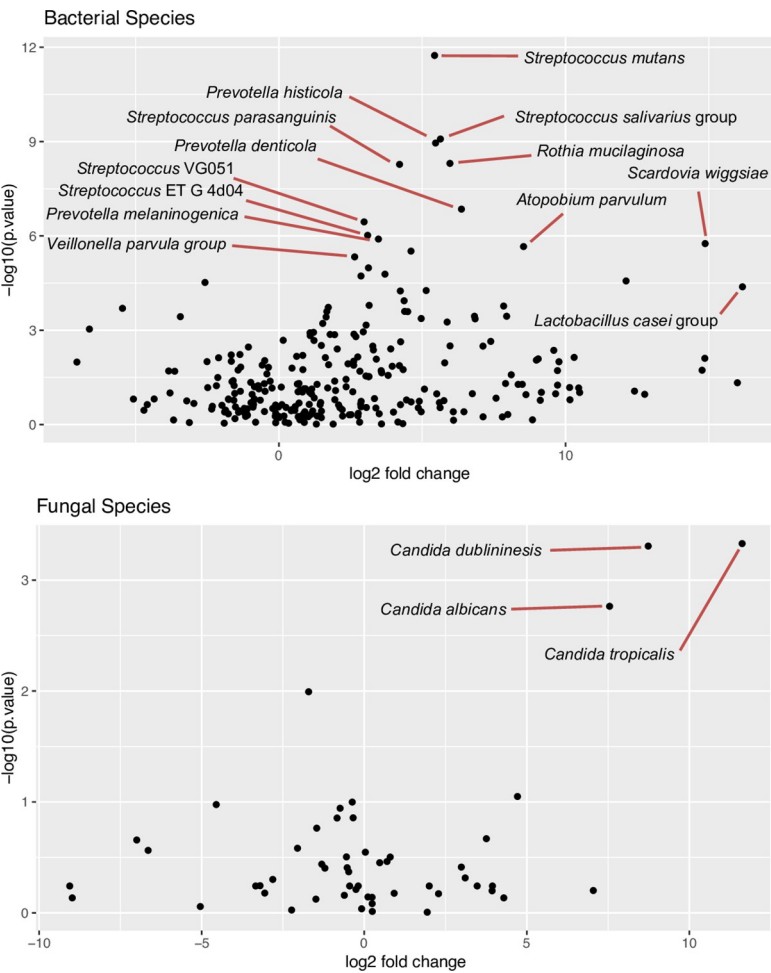

**Fig 3. Volcano plots of microbial species with the x axis representing the log base 2 of the fold change between dentin lesions and intact enamel in estimated absolute abundance and the y axis representing the negative log base 10 of the p value from a Friedman test on the 4 sample types from caries-affected subjects.** Selected species are labeled. A) Bacterial species B) Fungal species.

One noticeable feature is that although *C. tropicalis* has a high mean abundance, that is almost completely due to one sample and that we would expect to observe *C. albicans* and *dubliniensis* histologically more often. S3C Fig shows a correlation plot of the fungal and bacterial species and we see only one interaction that is significantly different from random among the three major *Candida* species, a negative interaction between the *Veillonella parvula* group and *C. dubliniensis*.

We also did an analysis with the program DADA2 [61] using the ITS pipeline of the program to generate amplicon sequence variants (ASVs) of the fungal ITS2 data. We found that ASVs identified as *C. dubliniensis* were predominantly a single form (99.9%) with 5 rare ASVs. For *C. albicans* there was a predominant ASV and 297 less abundant ASVs. However, we observed minor ASVs at nearly the same abundance (around 26% of total) in the positive control (genomic DNA from a single strain of *C. albicans* mixed with other fungi) as in the experimental samples. It therefore was not possible to draw clear conclusions about the number of strains of either *C. albicans* or *C. dubliniensis* from ASV analysis.

As the microbiome analysis indicated the presence and predominance of fungi in carious dentin, we then examined for histologic evidence of this finding. Decalcified tissue sections of

primary teeth were stained with classical histochemical stains for fungi—Periodic acid-Schiff (PAS) stain, Grocott's Methenamine Silver (GMS), and calcofluor white (CW). On carious teeth, the magenta PAS stain highlighted fungal cell walls within the plaque biofilm located along the carious margins (Fig 4A). The PAS stain also revealed fungal extensions into the dentin tubules (Fig 4A and 4B). In contrast, no PAS staining was observed in intact dentin of non-carious teeth (Fig 4C). The GMS stain, which precipitates silver ions in fungal polysaccharide walls, showed a similar pattern of staining in the plaque and dentinal tubules proximal to carious lesions (Fig 4D and 4E), but GMS staining was absent in intact non-carious teeth (Fig 4F). Fungal extensions in tissue stained with both PAS and GMS traversed from the plaque to the interface between the primary dentin and the region of the pulp and tertiary dentin (Fig 4A and 4D). At 100X magnification, fungi can be seen completely occupying the width of the tubules. Calcofluor white stain, which binds to chitin, similarly showed fungal accumulation along the carious margin and penetrating into the dentin tubules (Fig 4G and 4H). No CW staining was observed within the dentin tubules of non-carious teeth (Fig 4I).

To confirm the presence of fungi within the dentinal tubules of carious lesions, immunohistochemistry (IHC) was performed on carious and non-carious dentin. Fungal β-D-glucan antibody that specifically recognizes β-D-glucan, which is a polysaccharide present within the cell walls of *C. albicans*, was multiplexed with a lipoteichoic acid (LTA) antibody to recognize this constituent of gram-positive bacterial cell walls. Multiplexed IHC using the β-D-glucan and LTA antibodies revealed the presence of fungi and gram-positive bacteria within the carious dentin tubules (Fig 5A and 5E). In most instances, individual tubules contained either fungi or gram-positive bacteria but these kingdoms rarely co-occurred (Fig 5A and 5E). No β-D-glucan or LTA staining was observed in the dentinal tubules of control intact, non-carious teeth (Fig 5I). Multiplexed IHC with β-D-glucan and gram-negative endotoxin revealed that fungi and gram-negative bacteria tend to segregate into separate tubules (S4A Fig), similar to what was observed with fungi and gram-positive bacteria (Fig 5A and 5E). Multiplexed IHC with gram-positive LTA and gram-negative endotoxin revealed that gram-positive and gram-negative bacteria also occupy separate tubules (S4B Fig). Correlative SEM was used to evaluate the spatial and morphologic structures of bacteria and fungi within carious and non-carious dentin (Fig 5C, 5D, 5G, 5H, 5K and 5L). Within carious dentin tubules, monomorphic bacterial and fungal biofilms existed in distinct tubules. Tubules immunolcalized for gram-positive bacteria demonstrated cocci or filamentous structures on SEM (Fig 5C and 5H). Similarly, tubules immunolocalized for β-D-glucan on IHC demonstrated fungal-like microscopic structures (Fig 5D and 5G). Empty dentin tubules (Fig 5K) or dentin tubules containing odontoblastic processes (Fig 5L) were observed in non-carious teeth.

While the multiplex staining with β-D-glucan and LTA antibodies provided a general overview of the distribution of fungi and gram-positive bacteria within the dentinal tubules, we also assessed the localization of representative species from each kingdom with disease relevance: *C. albicans* and *S. mutans*. This was accomplished by performing multiplex IHC with polyclonal antibodies directed against *C. albicans* and *S. mutans*. As with the more general staining for β-D-glucan and LTA, the *C. albicans* and *S. mutans* multiplex IHC revealed the presence of both organisms in carious dentin tubules (Fig 5B and 5F). In general, tubules containing these organisms tended to contain predominantly one or the other but not both (Fig 5B and 5E). The dentinal tubules of control non-carious teeth did not stain for *C. albicans* or *S. mutans* (Fig 5J).

To further confirm the distinct spatial occupation of fungi and gram-positive bacteria, we performed multiplex immunofluorescence (IF) staining by using the same antibodies used for IHC staining. Control non-carious teeth showed no staining within dentinal tubules (Fig 6A–6C), but carious dentinal tubules demonstrated staining with LTA (Fig 6D and 6G) or β-D-

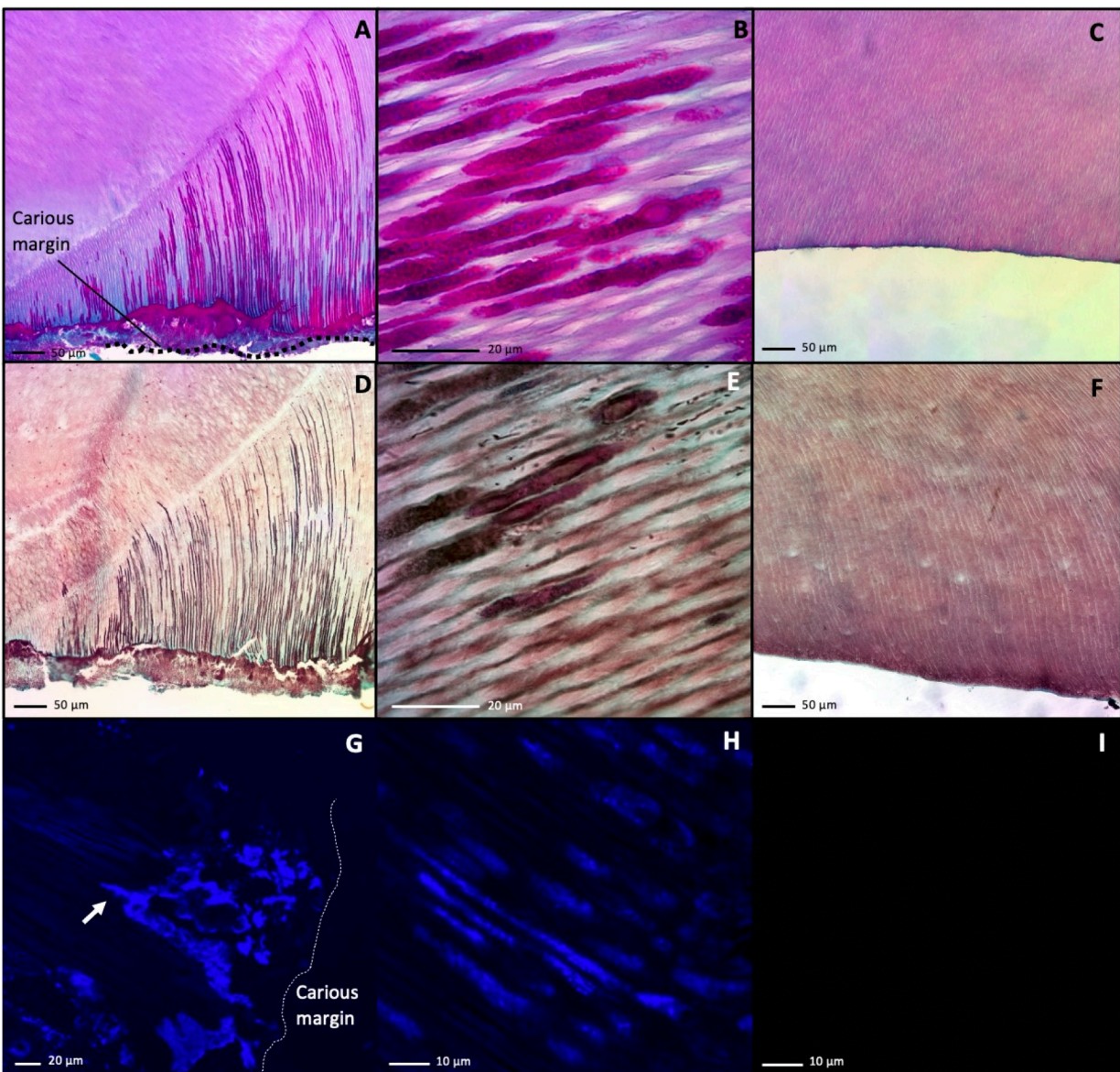

**Fig 4. Fungi Are Present Within the Dentinal Tubules of Carious Dentin But Not Non-Carious Dentin.** Histologic sections of carious or non-carious primary were stained with classic histologic stains to visualize fungi. (A) PAS-stained carious tooth section imaged via light microscopy using a 40X objective showing fungal elements (deep magenta) are seen penetrating dentinal tubules from the carious margin (bottom of image) towards the pulp (top of image). (B) PAS-stained carious tooth section from A visualized at higher magnification with a 100X objective demonstrates fungal elements (deep magenta) in dentinal tubules. (C) PAS-stained non-carious tooth section imaged via light microscopy using a 40X objective showed no fungal staining in dentinal tubules. (D) GMS stain from the same sample as in A imaged via light microscopy with a 40X objective showed a similar staining pattern as in A, with fungi stained a dark brown. (E) GMS-stained carious tooth section from B visualized at higher magnification with a 100X objective demonstrates fungal elements (dark) in dentinal tubules. (F) GMS-stained non-carious tooth section imaged via light microscopy using a 40X objective showed no fungal staining in dentinal tubules. (G) Calcofluor white stain of a carious tooth section imaged via fluorescent microscopy at 405 nm using a 40X objective showed fungal accumulation (blue) at the carious margin and starting to penetrate into the dentin tubules (white arrow). (H) Calcofluor white-stained carious tooth section from C visualized at higher magnification with a 100X objective demonstrates fungal elements (blue) in dentinal tubules. (I) Calcofluor white-stained non-carious tooth section imaged via fluorescent microscopy using a 40X objective showed no fungal staining in dentinal tubules. Scale bars are indicated in each panel.

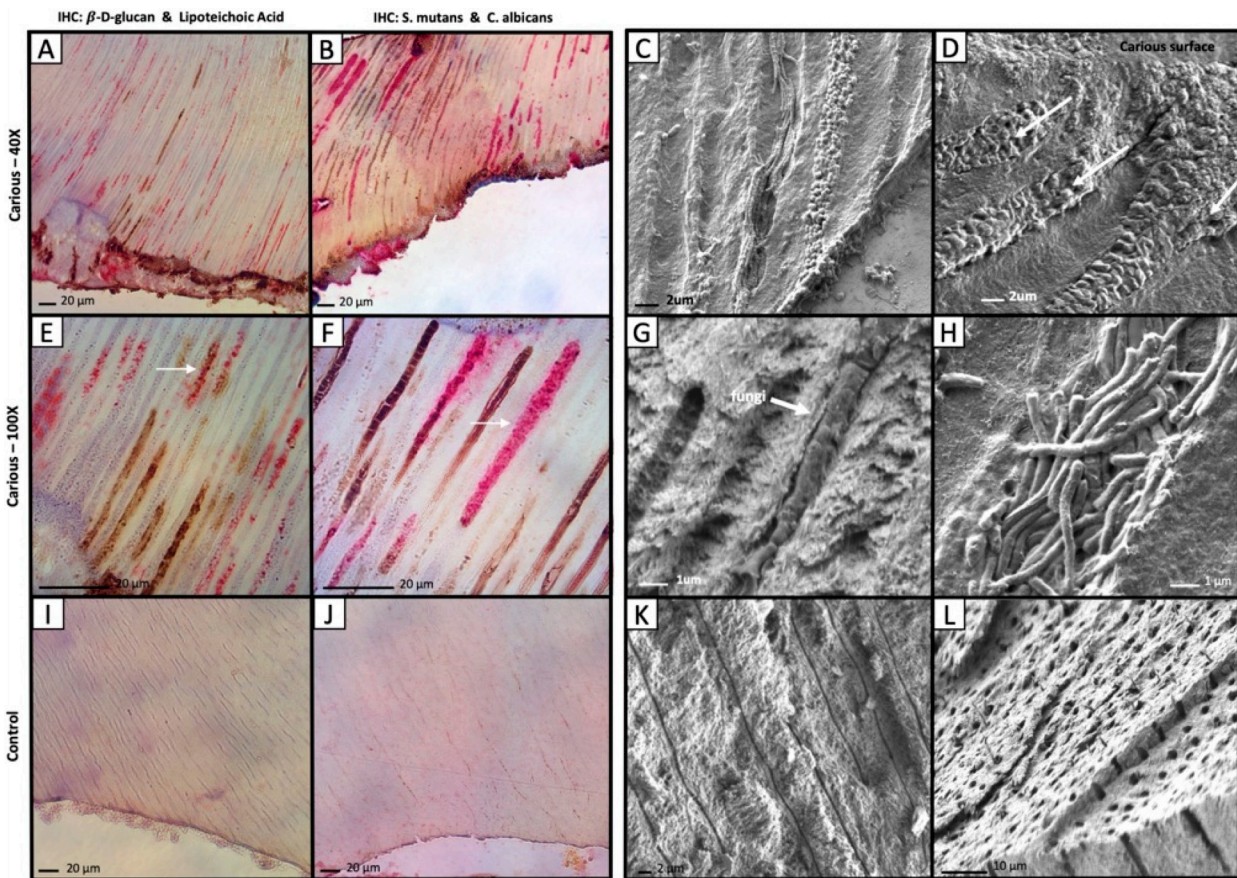

**Fig 5. Fungi and Bacteria Invade Carious Dentin and Occupy Distinct Tubules.** Carious or non-carious primary teeth were analyzed by IHC or SEM to visualize fungi and bacteria. (A) IHC with β-D-glucan (DAB, brown) and lipoteichoic acid (AP, magenta) of carious dentin at 40X magnification. (B) IHC with S. mutans (AP, magenta) and C. albicans (DAB, brown) at 40X magnification. (C) SEM image of carious dentin demonstrating filamentous bacteria and cocci in distinct dentinal tubules which immunolocalized lipoteichoic acid. (D) SEM image of carious dentin demonstrating tubules invaded with fungal elements which immunolocalized β-D-glucan (arrows). (E) IHC with β-D-glucan (DAB, brown) and lipoteichoic acid (AP, magenta) of carious dentin at 100X magnification. Arrow demonstrates a rare tubule where bacteria and fungi co-localized. (F) IHC with S. mutans (AP, magenta) and C. albicans (DAB, brown) at 100X magnification. Arrow demonstrates cocci-shaped bacteria infiltrating the tubule. (G) SEM image of carious dentin demonstrating fungi in a dentinal tubule. (H) SEM image of carious dentin demonstrating filamentous bacteria in a dentin tubule. (I) IHC with β-D-glucan (DAB, brown) and lipoteichoic acid (AP, magenta) of healthy control dentin at 40X magnification demonstrating no staining within the dentinal tubules. (J) IHC with S. mutans (AP, magenta) and C. albicans (DAB, brown) at 40X magnification demonstrating no staining within the dentinal tubules. (K) SEM image of healthy control dentin demonstrating empty dentin tubules. (L) SEM image of healthy control dentin demonstrating dentin tubules containing odontoblast processes. Scale bars are indicated in each panel.

glucan (Fig 6E and 6H), as previously observed with IHC. The IF staining also redemonstrated the predominance of fungi or gram-positive bacteria in individual tubules (Fig 6F and 6I). A mixed population of both fungi or gram-positive bacteria within the same tubules was occasionally observed (Fig 6I).

A similar approach was taken with multiplex IF using *C. albicans* and *S. mutans* antibodies (Fig 6J–6R). The dentinal tubules of control non-carious teeth did not demonstrate IF staining with this antibody combination (Fig 6J–6L) in agreement with the observation by IHC. Overall, individual dentinal tubules within carious lesions contained either predominantly *S. mutans* (Fig 6M and 6P) or *C. albicans* (Fig 6N and 6Q), though in some cases a mixture was observed (Fig 6R).

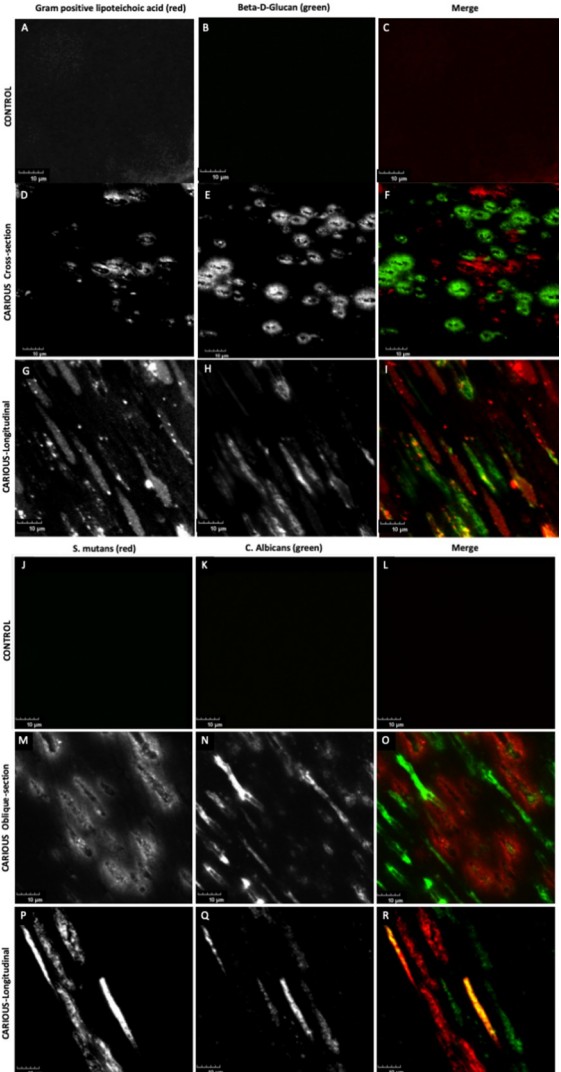

**Fig 6. Fungi and Bacteria Rarely Colocalize the Same Dentinal Tubules in Carious Dentin.** Multiplex immunofluorescence imaging of non-carious primary teeth revealed no gram-positive lipoteichoic acid, indicative of bacteria (A) or β-D-glucan, indicative of fungi (B). Merged images of A and B shown in C indicate no signal. Carious primary teeth visualized in cross-section revealed the presence of gram-positive lipoteichoic acid (D) and β-D-glucan (E). Merged image of D and E is shown in F. (F) Gram positive bacteria (red) and fungi (green) generally occupied distinct dentinal tubules with rare instances of colocalization within the same tubules. Carious primary teeth visualized in longitudinal-section revealed the presence of gram-positive lipoteichoic acid (G) and β-D-glucan (H). Merged image of G and H is shown in I. (I) Gram positive bacteria (red) and fungi (green) generally occupied distinct dentinal tubules with rare instances of colocalization (yellow in merge) within the same tubules. Multiplex immunofluorescence of non-carious primary teeth revealed no *S. mutans* (J) or *C. albicans* (K). Merged images of J and K shown in L indicate no signal. Carious primary teeth visualized in oblique section revealed the presence of *S. mutans* (M) and *C. albicans* (N), Merged image of M and N is shown in O. (O) *S. mutans* (red) and *C. albicans* (green) generally occupied distinct dentinal tubules with rare instances of colocalization within the same tubules. Carious primary teeth visualized in longitudinal-section revealed the presence of *S. mutans* (P) and *C. albicans* (Q). Merged image of P and Q is shown in R. (R) *S. mutans* (red) and *C. albicans* (green) generally occupied distinct dentinal tubules with rare instances of colocalization (yellow in merge) within the same tubules. Scale bar represents 10 μm.

## Discussion

This study couples DNA-based microbial community analysis of bacteria and fungi with high resolution imaging to explore the microbial invasion of dentin in deep caries. Our current understanding of caries pathophysiology has largely focused on bacteria in surface biofilms, although recently, the role of bacterial-fungal interactions in plaque has been recognized. We have observed both bacteria and fungi to be ubiquitously present in carious dentin but absent in healthy dentin. We also found exclusion, in that fungi and bacteria occupy separate dentinal tubules with rare instances of co-colonization within the same tubule, and that bacterial morphotypes vary among tubules. These phenomena are opposite to what is observed on the surface biofilm, where cross-kingdom communities contain bacteria and fungi that are intermixed and co-adhered to one another. The identification of fungi in dentin is particularly novel, and there is indication of fungal involvement in deeper infections from prior literature. Fungi are found in at least ten percent of all endodontic infections,[62] and is likely to be much higher in ECC patients. Currently, antibiotics are the first-line therapy for endodontic and odontogenic infections.[63, 64] Antibiotic treatment of chronic odontogenic infections is likely to be ineffective against fungi, resulting in the potential spread of systemic fungal infections. Thus, the findings of this study impact our current treatment of odontogenic infections and warrants further study.

The amplicon sequencing of bacteria was largely consistent with previous studies [13,25,28]. *S. mutans* was particularly associated with caries, along with multiple other bacteria such as the *Veillonella parvula* group, *Scardovia wiggsiae*, the *Streptococcus salivarius* group, and *Streptococcus parasanguinis*. All of these have shown significant association with caries in previous studies [13,25].

Fungal biodiversity within carious dentin was lower relative to bacteria. Fungal community profiling showed both *C. albicans* and the closely related *C. dubliniensis* to be strongly associated with caries and present high numbers within carious dentin. A single fungal species was almost always dominant within an individual carious tooth; co-colonizing fungi of distinct species was seldom observed. Both *C. albicans* and *C. dubliniensis* have previously been associated with severe caries [28]. We also found high levels of *C. tropicalis* in a single subject, suggesting it may have the same pathogenic potential, but is less common.

Quantitative PCR measurement of total fungi indicated that the ratio of fungus to bacteria was higher in carious dentin than in surface biofilms associated with carious lesions, suggesting that fungi may assume an increasingly important role as caries progresses to invasion of the pulp. This is the serious, end stage of dental caries that leads to pain, pulpal death, and the spread of infection to adjoining tissues. Dentin presents an ideal niche for microbes due to its tubular morphology and rich nutritional content. The presence of fungi within this niche is an unexplored source for odontogenic infection with potentially dangerous systemic consequences. For this reason, invasion of dentin by *Candida* deserves further examination.

Fungi and bacteria predominantly occupied separate tubules within carious dentin, with occasional instances of co-occupation, observed by all imaging modalities used in this study. Individual tubules of carious dentin were occupied by low complexity, perhaps even single-species communities, and variation was observed among tubules, but not within. The invasion of dentin tubules by *Candida* has been previously observed *in vitro* and, in limited circumstances, *in vivo*. *C. albicans* hyphae have been shown to penetrate dentin tubules in laboratory experiments [56,58,65,66], demonstrating a potential pathway of infection. Case reports have demonstrated dentinal candidiasis in teeth lacking enamel [67], in immunocompromised individuals [68], and in carious dentin from HIV-positive children [55] Our study extends these earlier findings by suggesting that this is a general phenomenon in ECC. In this study, our

qualitative histologic analyses were based on a limited sample size of 20. While histologic findings were congruent amongst the teeth sampled, histologic assessment of teeth with varying degrees of caries disease would provide further insight on the inter-kingdom interactions between bacteria and fungi that result in the advancement of caries in future studies.

The separation of bacteria and fungi in distinct tubules may be attributable to factors such as the limited diameter of the tubule, limited physical interactions between *C. albicans* and S. *mutans*, or antagonistic interactions between *C. albicans* and *S. mutans*, which have been observed in some environments. While some physical interactions between some *Streptococcus* species and *C. albicans* have been observed, the physical interactions between *S. mutans* and *C. albicans* are less robust than several other *Streptococcus* species [69]. The observed separation between *S. mutans* and *C. albicans* in dentinal tubules may be a reflection in part of the fact that they are less likely to physically interact and therefore less likely to be in close physical proximity. Notably, glucose and the presence of *S. mutans* suppresses hyphal formation in *C. albicans* [70]. Since carbohydrates help mediate this interspecies interaction, the reduced carbohydrate load in dentinal tubules [19,71] may also explain the reduced co-localization of these species within individual dentinal tubules. C. *albicans* has been shown to have a bacteriostatic effect against *S. mutans* [72]. Quorum sensing molecules secreted by *S. mutans* can inhibit hyphae and fungal biofilm development [70,73–75], attenuating the virulence of *C. albicans*. Dual species biofilms of *C. albicans* and *S. mutans* grown on hydroxyapatite disks exhibited higher pH of the growth environment and less demineralization compared to single species biofilms [76]. Antagonistic cross-kingdom relationships have also been reported between *C. albicans* and *P. aeruginosa* [77] and *Acinetobacter baumannii* [78]. However, the cross-kingdom relationship may not be fully antagonistic, as some tubules contained both *S. mutans* and *C. albicans*. The robust demineralizing activity of *S. mutans* and the dentinophilic characteristics of *C. albicans* may suggest synergism across the two phases of caries pathophysiology–enamel demineralization and dentin degradation. The presence of *S. mutans* to initiate an enamel defect that allows exposure of dentinal tubules may be required for subsequent *Candidal* invasion of the dentinal tubules. Cariogenic bacteria break down fermentable carbohydrates and create an acidogenic environment in the process, resulting in demineralization and formation of a carious lesion [79].

Carious dentin could represent a unique niche for multi-kingdom microbial growth. Despite an abundance of studies on bacterial and bacterial-fungal degradation of enamel, few studies have specifically linked proteolytic activity of bacteria to the degradation of dentinal collagen. Only in recent literature has the proteolytic activity of *S. mutans* been demonstrated, and only in *in vitro* models [80,81]. Some proteolytic bacteria such as *Prevotella* and *Propionibacterium* have been identified in carious dentin [53] and root surfaces [82,83] by DNA-based studies, but dentin degradation by these microbes has not been demonstrated. This suggests that bacterial partnership with other microbial kingdoms capable of proteolysis may be important in caries progression. The highly collagenous composition of dentin provides an ideal niche for *Candida*, which is effective at binding, metabolizing, and degrading collagen [84,85]. Furthermore, dentinal fluid rich in immunoglobulins and albumin serves as a nutrient source for fungi [19]. *Candida* has been described as a dentinophilic microorganism, with thigmotropic behavior that allows directional growth as a mechanosensory response to contact [57]. Demineralization-induced exposure of dentin tubules, which accommodate the size of *Candida*, may provide an open surface for initial fungal entry. Once it has invaded dentinal tissues, *Candida* possesses properties that allow it to exist indefinitely, effectively protected, and inaccessible to antimicrobial treatments. This niche renders carious dentin as a constant source of pathogenic fungi, which has important implications for the perpetuation of dental caries and its management. The cross-kingdom interactions between *C. albicans* and *S. mutans* are

complex and were not addressed in this study but require further investigation, especially in carious dentin.

On SEM, bacteria tended to colonize with other species of the same morphology within individual dentin tubules. Cocci and filaments were the most prevalent morphotypes. Specifically, filamentous bacteria colonized distinct tubules from cocci bacteria. A wide spectrum of microbial traits influence biofilm formation, including initial cell adhesion, production of extracellular polysaccharides, quorum sensing, motility, and cell metabolism [86]. This phenomenon of single genera forming spatial microenvironments has been observed in other porous microenvironments [87]. This model is speculated to occur because individual genera compete and proliferate more rapidly in microenvironments that support their physiological demands. The anatomic and physiologic niches of dentinal tubules are conducive to specific genera invading more effectively. Filamentous bacteria can thrive in tubules with nutrient-limiting conditions because their higher surface area allows for better contact with nutrients [88], whereas cocci may have better access to smaller tubules. Overall, microbial organization in dentin is influenced by the microenvironment of individual dentin tubules, which is determined by factors such as tubule diameter and the composition of dentinal fluid [89].

This study showed a higher ratio of fungi to bacteria in carious dentin than in surface biofilms, suggesting an underappreciated role for fungi in invasive caries. It also identified distinctive spatial behavior in polymicrobial microbial invasion of dentin that is unlike the polymicrobial structure of the well-characterized surface biofilm. Further exploration of the spatial relationship of fungal and bacterial communities within carious dentin and their functional interactions are needed to better understand the pathophysiology of invasive caries. This knowledge may catalyze a paradigm shift in our understanding of invasive caries that could lead to improved preventive and therapeutic strategies.

## Materials and methods

### Ethics statement

All research involving human subjects was approved by institutional review boards. The subjects for microbiome analysis were recruited at the Nationwide Children's Hospital dental clinics under a protocol approved by the Nationwide Children's Hospital Institutional Review Board (protocol # NCH 00000478). The subjects for imaging analysis were recruited at the Department of Dentistry at Boston Children's Hospital using a protocol approved by the Boston Children's Hospital Institutional Review Board (IRB-P00030469). In all cases formal written consent was obtained from the parents or guardians.

### Subjects

The subjects for microbiome analysis at Nationwide Children's Hospital ranged in age from 24–59 months old with a median of 48. The subjects for imaging analysis at Boston Children's Hospital were normal healthy children (American Society of Anesthesiologist Classification System I [ASA I]) under the age of 10 years seen in the Department of Dentistry at Boston Children's Hospital requiring extraction of primary teeth. Carious primary teeth (including molars, incisors, and canines) were extracted from ten subjects with varying severity of caries due to abscess or imminent exfoliation. Control primary teeth (including molars, incisors, and canines) were extracted from ten subjects due to orthodonic indications or imminent exfoliation. Caries classification at both study sites was based on the International Caries Detection and Assessment System [90].

## Sample preparation and sequencing

Plaque was collected with microbrushes from intact enamel in caries-free control subjects (n = 43) and from intact enamel and carious lesions from caries-affected subjects (n = 32), namely white spots and cavitated lesions. A spoon excavator was used to collect carious dentin from dentin-involved lesions. The microbrushes were placed into 0.3 ml of buffer ATL and stored at -20˚C until processing. DNA was prepared by the QIAamp DNA blood mini kit (Qiagen, USA) with the addition of an initial bead-beating step using 0.1 mm beads for bacteria and 0.5 mm beads for fungi (0.25 g beads, BioSpec Products BeadBeater).

Sequencing libraries were prepared with two amplification steps as described in the Illumina 16S Metagenomic Sequencing Library Preparation document. The primary amplification was carried out with primers designed to amplify the V1V3 region of the 16S rRNA gene for bacterial identification [91] or the ITS2 region of the fungal rRNA operon [92] Modifications were made to the secondary amplification step to introduce additional barcodes [93]. Sequencing was carried out on the MiSeq system with 2 x 300 bp reads which allows approximately a 50 bp overlap for the 16S amplicons and variable longer overlap for the ITS2 amplicons.

## Quantitative PCR

Quantitative PCR (qPCR) for total bacterial and total fungal DNA was carried out as previously described [92]. Briefly the primers used were from the 16S rRNA gene for bacteria and similar primers as were used for the ITS2 amplification for fungi. *Porphyromonas gingivalis* genomic DNA and *Candida albicans* genomic DNA were used as standards and the quantitation was expressed as *P. gingivalis* genome equivalents per microliter or *C. albicans* genome equivalents per microliter respectively. There are two rationales for using *P. gingivalis* genomic DNA as standard. One is that we had used it in previous studies [94], and it allowed for standardization and quality control in our laboratory. Secondly *P. gingivalis* has a medium density of ribosomal RNA operons with 4 in a 2.3 Mbp genome. We note that measurement of bacterial numbers by qPCR in a metagenome will necessarily be an estimate.

## Sequence data analysis

Forward and reverse reads were aligned with the mothur function make-contigs, mothur screen-seqs was used to select contigs with less than 10 ambiguous bases and of restricted lengths (greater than 450 bp for 16S and greater than 150 for ITS2), and mothur trim.seqs was used to remove primer sequences [95,96]. The contigs were aligned with the CORE oral database [97] and were classified as the highest percentage match greater than 98% identity. To generate estimates of absolute abundance of fungal and bacterial species, the qPCR quantitation for the entire sample was multiplied by the fraction of the sample for each species determined by sequence counts.

## Statistics and data visualization

To determine if the ratio of fungal to bacterial DNA changed significantly with severity of carious lesion, we applied a linear mixed effects model to the log-transformed ratio data using the lmer function of the lme4 library [98] in R with subject used as the random effect. The test for significance that was used was the Anova function of the car library in R, which implements a type II Wald chi square test. To determine the significance of differences in the overall microbial communities between intact enamel and various lesions we used a PERMANOVA test implemented by the adonis function of the vegan library in R [99]. The communities were

visualized in non-metric multidimensional scaling using the metaMDS function of the vegan library in R. To determine whether individual microbial species varied significantly between healthy enamel and different lesion severity in caries subjects we applied the Friedman test using subject as the block (function friedman.test in base R). Spearman correlations between bacterial and fungal species were determined with the rcorr function of the Hmisc library in R and plotted with the ggcorrplot function of the eponymous library in R. Analysis code for this project is available at https://github.com/cliffbeall/dentin_caries.

## Histologic tissue processing

Extracted teeth were placed in 10% neutral-buffered formalin immediately following extraction. After 24 hours in formalin, the teeth were transferred to Christensen's buffer (containing formic acid) for decalcification at room temperature for 14 days. Following fixation and decalcification, samples were processed for paraffin embedding using Leica ASP300S tissue processor (Leica Biosystems). Sections were cut at 5 μm thickness on a Leica RM2255 rotary microtome and mounted on Fisherbrand Superfrost Plus microscope slides (12-550-15) for subsequent staining.

## Antibodies

The antibodies used in this study (Table 1) were purchased from commercial suppliers and stored according to manufacturer instructions prior to use.

## Periodic acid-Schiff stain

Periodic Acid Schiff Reaction (PAS) staining was performed using the Periodic Acid Schiff Reaction (PAS) Kit (Poly Scientific R&D Corp, #k047) following the staining procedure outlined in the kit manual. In brief, slides were deparaffinized and hydrated to water. Slides were stained with Alcian Blue 1% in 3% acetic acid (pH 2.5) for 5 minutes and then rinsed in distilled water. The slides were dipped in periodic acid 1% aqueous for 2 minutes followed by a rinse in distilled water. Staining with Schiff reagent was performed for 8 minutes, and the slides were subsequently rinsed in running water for 5–10 minutes. Slides were counterstained with Mayer's Modified Hematoxylin for 2 minutes and then rinsed in distilled water. The slides were differentiated with 3–4 dips in acid alcohol (0.5% aqueous) and rinsed with distilled water. The slides were dehydrated in absolute alcohol and cleared in xylenes. The coverslips were then mounted.

## Grocott's methenamine silver stain

Grocott's methenamine silver (GMS) staining was performed using the Grocott's method for Fungi (GMS) kit (Poly Scientific R&D Corp, #k022) following the staining procedure outlined in the kit manual. In brief, slides were deparaffinized and then hydrated to distilled water. They were then oxidized in chromic acid 4% aqueous for 1 hour followed by a brief wash in tap water. The slides were placed in sodium bisulfite 1% aqueous for 1 minute and then washed in running water for 5–10 minutes. They were subsequently rinsed in 3–4 changes of distilled water. The slide were then placed in freshly mixed methenamine silver nitrate working solution and incubated in an oven at 58–60°C for 60 minutes (or until yellowish brown). The slides were rinsed in 6 changes of distilled water and then toned in gold chloride 0.1% aqueous for 2–5 minutes. They were subsequently rinsed in distilled water. Unreduced silver was removed with sodium thiosulfate 2% aqueous for 2–5 minutes and then washed thoroughly in tap water. The slides were counterstained with Fast Green substitute for Light Green 0.2% aqueous

**Table 1. Primary and Secondary Antibodies.**

| Antibody | Host/Isotype | Manufacturer | Catalog # | Dilution |
|---|---|---|---|---|
| Fungal Beta Glucan | Rabbit/IgG1 | ThermoFisher | MA5-33305 | 1:50 |
| Gram Positive Bacteria LTA (lipoteichoic acid) | Mouse/IgG1 | ThermoFisher | MA1-7402 | 1:100 |
| Gram Negative Endotoxin | Mouse/IgM | ThermoFisher | MA1-10692 | 1:100 |
| Streptococcus mutans | Rabbit/polyclonal | Antibodies-online.com | ABIN4888516 | 1:100 |
| Candida albicans | Rabbit/polyclonal | Abcam | ab53891 | 1:2000 |
| Secondary antibody | | Leica–Refine Polymer Detection Kit | DS9800 | |

for 30–45 seconds. The slides were dehydrated in 95% alcohol and absolute alcohol for 2 changes each followed by clearing in xylenes for 2 changes each. The coverslips were then mounted and the slides were visualized using an Olympus IX81 inverted microscope.

## Calcofluor white stain

Staining was performed using Calcoflluor White Stain (Sigma-Aldrich 18909) following the procedure outlined in the kit manual. In brief, slides were deparaffinized and then hydrated to distilled water. One drop off CW and one drop of 10% potassium hydroxide solution was added to the sample. The coverslips were then mounted and visualized using an Olympus Fluoview FV1000 Confocal microscope.

## Immunohistochemistry

Single immunohistochemistry was performed on the Leica Bond III automated staining platform using the Leica Biosystems Refine Detection Kit (DS9800). All antibodies were diluted in Leica Primary Antibody Diluent (AR93520). The primary antibodies listed in Table 1 were used at the dilutions indicated. No antigen retrieval was used for any of the primary antibodies.

Dual chromogenic IHC Immunohistochemistry was performed sequentially on the Leica Bond III automated staining platform using the Leica Biosystems Refine Detection Kit (DS9800) and Leica Biosystems Refine Detection Kit (DS9390). All antibodies were diluted in Leica Primary Antibody Diluent (AR93520). The primary antibodies listed in Table 1 were used at the dilutions indicated. For one pair of antibodies, *Candida albicans* was visualized via 3, 3'-diaminobenzidine (DAB) and *S. mutans* was visualized via alkaline phosphatase (AP). For another antibody pair, fungal β-D glucan was visualized via DAB and gram-positive bacteria LTA was visualized via AP. For a third antibody pair, gram-negative endotoxin was visualized via DAB and gram-positive bacteria LTA was visualized via AP. For a fourth antibody pair, β-D glucan was visualized via DAB and gram-negative endotoxin was visualized via AP. The coverslips were then mounted and the slides were visualized using an Olympus IX81 inverted microscope.

## Immunofluorescence

Dual immunofluorescence staining was performed sequentially on the Leica Bond III automated staining platform using the Leica Biosystems Refine Detection Kit (DS9800). All antibodies were diluted in Leica Primary Antibody Diluent (AR93520). For one pair of antibodies, *Candida albicans* was visualized via Alexa Fluor 647 (Life Technologies; B40958) and *S. mutans* was visualized via Alexa Fluor 555 (Life Technologies; B40955. For another antibody pair, fungal β-D glucan was visualized via Alexa Fluor 647 (Life Technologies; B40958) and gram-positive bacteria LTA was visualized via Alexa Fluor 647 (Life Technologies; B40958). For a third antibody pair, gram-negative endotoxin was visualized via Alexa Fluor 647 (Life

Technologies; B40958) and gram-positive bacteria LTA was visualized via Alexa Fluor 555 (Life Technologies; B40955). The coverslips were then mounted and visualized using an Olympus Fluoview FV1000 Confocal microscope.

## Scanning electron microscopy

Whole teeth were cryofractured and histologic sections that were immunostained for β-D-glucan and LTA were gold-coated and viewed utilizing field emission scanning electron microscopy (FESEM); SIGMA VP500 Carl Zeiss Microscopy at 1 keV under various magnifications.

## Supporting information

**S1 Fig. Non-metric multidimensional scaling of Bray-Curtis dissimilarities of relative abundance data for microbes from different lesion type samples as shown in the legend.** The black arrows show the position of centroids of the various groups and the ellipses show 95% confidence intervals of the centroind position. (A) Bacterial samples (B) Fungal Samples. (TIF)

**S2 Fig. Bubble plots of the identical NMDS plots as in Fig 2 but with points sized by the estimated absolute abundance of microbial species.** (A) Fungal NMDS sized by *Candida albicans* abundance (B) Fungal NMDS sized by *Candida dubliniensis* abundance (C) Bacterial NMDS sized by *Streptococcus mutans* abundance. (TIF)

**S3 Fig. Abundance and correlations of the most abundant bacteria and fungi in dentin lesions.** (A) Dotplot of the estimated absolute abundance of 15 bacterial species with the highest mean abundance across dentin lesion samples. (B) Dot plot of the estimated absolute abundance of the 5 fungal species with highest mean abundance across dentin samples. (C) Spearman correlations between the bacterial and fungal groups. Color indicates the strength of correlation and the x's indicate correlations that are not significantly different than random. (TIF)

**S4 Fig. Gram-Negative Bacteria, Gram-Positive Bacteria, and Fungi Occupy Distinct Dentin Tubules.** Carious primary teeth were analyzed by IHC to visualize fungi and bacteria. (A) IHC with gram-negative endotoxin (DAB, brown) and lipoteichoic acid (AP, magenta) of carious dentin at 40X magnification. (B) IHC with β-D-glucan (AP, magenta) and Cgram-negative endotoxin (DAB, brown) at 40X magnification. Scale bars are indicated in each panel. (TIF)

**S1 Table. PERMANOVA results for bacterial and fungal communities in intact enamel and various lesion types.** $R^2$ values are in the upper right and p values in the lower left. Based on Bray-Curtis dissimilarity of estimated absolute abundance of species incorporating qPCR data. (DOCX)

**S2 Table. Results of Friedman tests for variation in estimated absolute abundance for bacterial species based on combined QPCR and 16S rRNA gene amplicon sequencing.** (XLSX)

**S3 Table. Results of Friedman tests for variation in estimated absolute abundance for fungal species based on combined QPCR and ITS2 amplicon sequencing.** (XLSX)

## Acknowledgments

We thank Dana-Farber/Harvard Cancer Center in Boston, MA, for the use of the Specialized Histopathology Core, which provided histology and immunohistochemistry service. Dana-Farber/Harvard Cancer Center is supported in part by an NCI Cancer Center Support Grant # NIH 5 P30 CA06516.

We also thank the Boston Children's Hospital, Program in Cellular and Molecular Medicine Microscopy Core and Harry Leung for the use of the Olympus IX81 inverted microscope and Olympus FV 1000 Confocal System.

## Author Contributions

**Conceptualization:** Rosalyn M. Sulyanto, Clifford J. Beall, Kasey Ha, Joseph Montesano, Jason Juang, John R. Dickson, Eugene J. Leys, Mira Edgerton, Ann L. Griffen.

**Data curation:** Rosalyn M. Sulyanto, Clifford J. Beall.

**Formal analysis:** Rosalyn M. Sulyanto, Clifford J. Beall, Kasey Ha, Joseph Montesano, Jason Juang.

**Funding acquisition:** Rosalyn M. Sulyanto, Eugene J. Leys, Ann L. Griffen.

**Investigation:** Rosalyn M. Sulyanto, Kasey Ha, Joseph Montesano, Jason Juang, Shahr B. Hashmi, Seth Bradbury, Sunita P. Ho.

**Methodology:** Rosalyn M. Sulyanto, Clifford J. Beall, Kasey Ha, Joseph Montesano, Jason Juang, John R. Dickson, Ann L. Griffen.

**Project administration:** Rosalyn M. Sulyanto, Clifford J. Beall, Eugene J. Leys, Ann L. Griffen.

**Resources:** Rosalyn M. Sulyanto, Seth Bradbury, Eugene J. Leys, Ann L. Griffen.

**Software:** Clifford J. Beall.

**Supervision:** Rosalyn M. Sulyanto, Clifford J. Beall, Eugene J. Leys, Ann L. Griffen.

**Visualization:** Rosalyn M. Sulyanto, Clifford J. Beall, Kasey Ha, Joseph Montesano, Jason Juang, Sunita P. Ho, Ann L. Griffen.

**Writing – original draft:** Rosalyn M. Sulyanto, Clifford J. Beall, Kasey Ha, Joseph Montesano, Jason Juang, Ann L. Griffen.

**Writing – review & editing:** Rosalyn M. Sulyanto, Clifford J. Beall, John R. Dickson, Mira Edgerton, Sunita P. Ho, Ann L. Griffen.

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
