## [Decision Letter · Decision Letter 0]

28 Mar 2024

Dear Dr. Beall,

Thank you very much for submitting your manuscript "Fungi and Bacteria Occupy Distinct Spatial Niches within Carious Dentin" for consideration at PLOS Pathogens. As with all papers reviewed by the journal, your manuscript was reviewed by members of the editorial board and by several independent reviewers. The reviewers appreciated the attention to an important topic. Based on the reviews, we are likely to accept this manuscript for publication, providing that you modify the manuscript according to the review recommendations.

Sincerely,

Jason E Stajich, Ph.D.

Guest Editor

PLOS Pathogens

Alex Andrianopoulos

Section Editor

PLOS Pathogens

Michael Malim

Editor-in-Chief

PLOS Pathogens

orcid.org/0000-0002-7699-2064

Reviewer Comments (if any, and for reference):

Reviewer's Responses to Questions

**Part I - Summary**

Reviewer #1: Overall, this study is an important contribution to our understanding of caries.

Reviewer #2: This is an interesting and well-performed study which examines the microbiome and spatial distribution in dentin from Early Childhood Caries (ECC). S. mutans and Veillonella were associated with caries and there was also a higher ratio of fungi to bacteria in more severe lesions. The most notable finding was that fungi and streptococci primarily occupied separate dentinal tubules.

**Part II – Major Issues: Key Experiments Required for Acceptance**

Reviewer #1: Concerns:

1. First, plaque was collected from the surface of exposed dentin. It was not collected from within a tubule. Second, the histology omitted gram-negative bacteria. Consequently, we do not have a complete picture (taxonomic profile) within the tubules and fungi and bacteria may co-colonize tubules more frequently than reported. The environment within a tubule is likely distinct from the surface of dentin. For example, it could be more anaerobic. This would select for different bacteria. This distinction should be made clear. Specifically, that the dentin taxonomic profiles presented do not necessarily reflect the profile within a tubule. The situation is perhaps analogous to the transition between supragingival and subgingival plaque - each harbor distinct communities that contain both gram-positive and gram-negative bacteria.

2. Another issue the small sample size.

Points 1 and 2 are important caveats that need to be addressed and/or clearly stated at least in the abstract.

The distinction between minor and major revision hinges on how the authors wish to interpret and present their findings. For example, a major revision would be the addition of histology for gram-negative bacteria.

Reviewer #2: 1. Include a rationale for the use of P. gingivalis DNA as the standard for bacterial quantitation. It seems an unusual choice for a caries study. How does its rDNA gene load compare with that of the oral streptococci and other species that were identified in the study

2. Also include the rationale for collecting samples for microbiome or for imaging a two different sites and explain how clinical diagnosis was calibrated. Imminent exfoliation is given as a reason for extraction in both control and ECC patients.

**Part III – Minor Issues: Editorial and Data Presentation Modifications**

Reviewer #1: Minor point

L-278 "A single fungal species was almost always dominant within an individual carious tooth; co-colonizing fungi of distinct species was seldom observed."  For these teeth, were multiple ASVs of the same species observed? Did certain ASVs dominate? It maybe of interest to consider intra-specific variation. Considerable variation in biochemical characteristics among strains of a species may exist.

Reviewer #2: As S. mutans and C. albicans can be found intermixed and co-adhered to one another in the surface biofilm, none of the explanations offered for distinct dentinal tubule localization are entirely satisfactory. Some additional discussion would improve the paper.

PLOS authors have the option to publish the peer review history of their article (what does this mean?). If published, this will include your full peer review and any attached files.

Reviewer #1: No

Reviewer #2: No

Figure Files:

Data Requirements:

Reproducibility:

References:

---

## [Editor Report · Decision Letter 1]

9 May 2024

Dear Dr. Beall,

We are pleased to inform you that your manuscript 'Fungi and Bacteria Occupy Distinct Spatial Niches within Carious Dentin' has been provisionally accepted for publication in PLOS Pathogens.

Best regards,

Alex Andrianopoulos

Section Editor

PLOS Pathogens

Alex Andrianopoulos

Section Editor

PLOS Pathogens

Michael Malim

Editor-in-Chief

PLOS Pathogens

orcid.org/0000-0002-7699-2064
---

## [Editor Report · Acceptance letter]

16 May 2024

Dear Dr. Beall,

We are delighted to inform you that your manuscript, "Fungi and Bacteria Occupy Distinct Spatial Niches within Carious Dentin," has been formally accepted for publication in PLOS Pathogens.

Best regards,

Michael Malim

Editor-in-Chief

PLOS Pathogens

orcid.org/0000-0002-7699-2064